# Obese Vegetarians and Omnivores Show Different Metabolic Changes: Analysis of 1340 Individuals

**DOI:** 10.3390/nu14112204

**Published:** 2022-05-26

**Authors:** Eric Slywitch, Carine Savalli, Antonio Cláudio Duarte, Maria Arlete Meil Schimith Escrivão

**Affiliations:** 1Paulista School of Medicine, Federal University of São Paulo (Unifesp), Sao Paulo 04021-001, Brazil; maria.arlete@uol.com.br; 2Department of Public Policy and Public Health, Federal University of São Paulo (Unifesp), Sao Paulo 04021-001, Brazil; carine.savalli@unifesp.br; 3Medical School, Federal University of Rio de Janeiro (UFRJ), Rio de Janeiro 21941-901, Brazil; antonioclaudio@ufrj.br

**Keywords:** vegetarian diet, omnivorous diet, BMI, obesity, inflammation, insulin resistance, liver enzymes

## Abstract

Our study evaluated the association between the increase in body mass index (BMI) in men and women (menstruating and non-menstruating) (*n* = 1340) with different dietary groups (omnivores, semi-vegetarians, lacto-ovo-vegetarian, and vegans) and the measurement of the biochemical markers high-sensitive C-reactive protein (hs-CRP), ferritin, alanine aminotransferase (ALT), aspartate aminotransferase (AST), gamma-glutamyl transferase (GGT), glycated hemoglobin (HbA1C), and insulin resistance index (HOMA-IR). Increasing BMI values in all groups and dietary profiles were related to a significant increase in hs-CRP (*p* < 0.0001), ALT (*p* = 0.02), ferritin (*p* = 0.009), and HbA1C (*p* < 0.0001), with no difference between dietary groups (*p* < 0.05). The increase in BMI increases the levels of HOMA-IR (*p* < 0.0001) and GGT (*p* < 0.05), with higher values found in men when compared to women (*p* < 0.0001 for HOMA- IR and *p* = 0.0048 for GGT). The association between ALT and BMI was different between dietary groups, as it showed a decrease in vegan women who do not menstruate compared to other dietary groups (*p* = 0.0099). When including only obese individuals (BMI ≥ 30 kg/m^2^, *n* = 153) in the analysis, we observed lower concentrations of GGT and ferritin in vegetarians than in omnivores, regardless of gender and menstrual blood loss (*p* = 0.0395). Our data showed that for both vegetarians and omnivores, the higher the BMI, the worse the metabolic parameters. However, regarding obesity, vegetarians showed better antioxidant status (lower GGT elevation) and lower inflammatory status (lower ferritin elevation), which may provide them with potential protection in the development of morbidities associated with overweight.

## 1. Introduction

The increase in body mass index (BMI), due to increased adipose tissue, is associated with an increase in all-cause mortality [1].

Obesity leads to low-grade systemic inflammation with consequent insulin resistance and other metabolic disorders. Visceral fat, as well as that accumulated in the liver, heart, and muscles, increases the risk of type 2 diabetes mellitus (DM2), cardiovascular disease (CVD), asthma, and some types of cancer [2,3].

In this scenario, hepatic impairment is evident, with lipid accumulation in the liver proportional to the increase in BMI. Lipid accumulation can lead to nonalcoholic fatty liver disease (NAFLD), characterized by the involvement of more than 5% of hepatocytes infiltrated with fat, in the absence of diseases that compromise the liver, such as viral hepatitis, alcohol consumption, autoimmune hepatitis, and hereditary causes (hemochromatosis, Wilson’s disease). This condition can lead to nonalcoholic steatohepatitis (NASH), characterized by steatosis, inflammation, and liver damage, with or without fibrosis, which can progress to cirrhosis (requiring liver transplantation), hepatocellular carcinoma, and death. Hepatic inflammatory status appears to be the key factor in the progression from NAFLD to NASH [3,4].

NAFLD is the most common cause of the elevation of alanine aminotransferase (ALT) and aspartate aminotransferase (AST) transaminases, reflecting damage to hepatocytes [5]. Elevated ALT correlates with increased cardiovascular risk [6], as well as the risk of all-cause mortality [7].

In NAFLD, changes in the bile ducts are accompanied by a predominant increase in the enzyme gamma glutamyl transferase (GGT), the most sensitive marker of hepatobiliary disease, indicative of canalicular damage [5].

GGT is found in abundance on the luminal surface of proximal renal tubule cells. The liver has approximately one-fifth of the GGT present in the kidneys, and the greatest concentration is in the bile epithelial cells and bile canaliculi. Pancreatic acinar cells, but not islets, have GGT in their membrane, as do brain capillaries, astrocytes, reproductive system cells, and leukocytes [8].

In addition to being a marker of cholestasis, GGT also indicates the body’s oxidative state, as it participates in the synthesis of glutathione, the enzyme with the greatest antioxidant potential in mammals [9]. In the need for intracellular glutathione, GGT has its expression increased in the membrane, as it is able to cleave extracellular glutathione so that its amino acids (glutamic acid, cysteine, and glycine) can be transported to the intracellular space, where they become glutathione again [10]. As a decomposition product of glutathione, GGT produces cysteinyl glycine, a dipeptide that reacts with free iron, inducing the Fenton reaction, with the production of peroxide. Therefore, despite glutathione being a potent antioxidant, it has a pro-oxidant effect in the presence of GGT [11] and free iron, as illustrated in Figure 1.

Systematic reviews and meta-analyses associate increased circulating GGT with the risk of developing a variety of diseases involving metabolic inflammation and oxidative stress, independent of alcohol consumption and other factors. This risk is increased even when GGT values are within normal ranges [9,12,13,14,15,16,17,18,19]. Because it is an enzyme used for the conjugation of several compounds, GGT is considered a biomarker of environmental toxicity, due to contamination of xenobiotics [20]. It has an effect on the conjugation of xenobiotics at the blood–brain barrier and on the metabolism of vasoactive leukotrienes [8].

In addition, in insulin resistance and increased systemic inflammation, there is an increase in ferritin levels, which acts as a marker of inflammation, and not of the body’s iron reserve, as demonstrated in a previously published study [21]. The increase in circulating concentrations of this acute phase protein is a predictor of the development of metabolic syndrome and NAFLD in adults [22].

A well-planned vegetarian diet benefits the prevention and treatment of several noncommunicable chronic diseases, such as cardiovascular diseases, NAFLD, and diabetes, with better results when compared to well-planned omnivorous diets [23,24,25,26,27,28,29,30,31,32,33,34].

However, the literature has not demonstrated, to date, whether there are metabolic differences between vegetarians and omnivores with an increase in body mass index due to increased adipose tissue. The present study aimed to contribute with scientific information in this area, by analyzing the association between biochemical markers of inflammation, liver function, and insulin resistance and BMI values in vegetarian and omnivorous individuals.

## 2. Materials and Methods

### 2.1. Ethical Aspects

This retrospective, cross-sectional, and observational study was approved by the Research Ethics Committee of Paulista School of Medicine, Federal University of São Paulo (CEP 1.052.082). All participants of this study signed an Informed Consent.

### 2.2. Sample

The study included data from patients aged 18 to 60 years and of both sexes, treated at a private clinic in the city of São Paulo (Brazil) between 2008 and 2018. The clinic’s care profile focuses on metabolic and nutritional assessment, and most patients who seek it do not have serious diseases. Data from pregnant or lactating women were not included in the study; as well as patients using medications or supplements in the last six months, capable of influencing the variables evaluated; patients undergoing nutritional intervention in the last six months; smokers; patients with alcohol consumption more than three times a week; practitioners of physical activity more than three times a week; patients diagnosed with diseases capable of influencing the variables evaluated (thalassemia, liver diseases, hypo or hyperthyroidism, hemochromatosis, cancer with possible metabolic repercussions, rheumatologic and autoimmune diseases with a systemic inflammatory reflex, inflammatory bowel disease and irritable bowel syndrome) and gastrointestinal tract surgeries that affect nutrient absorption, such as total gastrectomy, partial or bypass, pancreatoduodenectomy, and bariatric surgery; eastern ethnicity and blood donors in the last 12 months. In the end, 1340 individuals were included, 422 men and 918 women, among which 226 do not menstruate and 691 have regular menstrual cycles (one of the women did not have information about menstrual cycle).

### 2.3. Sample Classification Regarding Eating Groups

The selected individuals were then classified into four groups, according to their food choices in the last 12 months: (1) omnivores (individuals who theoretically accept eating all food groups), (2) semi-vegetarians (individuals who eat white meats up to three meals a week), (3) lacto-ovo-vegetarians (individuals who do not consume any type of meat, but use eggs, milk, and dairy products), and (4) vegans (individuals who, in addition to abstaining from meat consumption, also do not use any animal products, such as eggs and dairy products). The grouping of semi-vegetarians, lacto-ovo vegetarians, and vegans into one group called “vegetarian” was later considered for comparative purposes with the omnivorous group.

### 2.4. Sample Classification Regarding Nutritional Status

The nutritional status of the individuals studied was determined from their BMI values, according to the criteria proposed by the World Health Organization [35]: low weight = BMI < 18.5; normal weight = BMI ≥ 18.5 and < 25.0, overweight = BMI ≥ 25.0 and < 30.0; and obesity = BMI ≥ 30.0 was obtained by dividing weight (in kg) by height squared (in meters). Body weight was measured with the patient barefoot in the center of a Filizola electronic scale with a capacity of 150 kg and accuracy of 100 g and wearing only underwear, while height was measured using an Alturexata stadiometer with an accuracy of 0.1 cm with the patient standing, barefoot, with heels together, spine erect, and arms extended alongside the body.

### 2.5. Analysis of Markers Associated with BMI

Serum concentrations of hs-CRP, liver enzymes (ALT, AST, and GGT), and ferritin, in addition to the glucose profile (fasting glucose, fasting insulinemia) and glycated hemoglobin (HbA1C), were evaluated. These biochemical concentrations were obtained from blood samples collected and analyzed in laboratories with international quality certification, chosen by the participants themselves. 

Values of hs-CRP concentrations were applied to assess the degree of inflammation, considering them as low (≤0.1 mg/dL), moderate (0.1 to 0.3 mg/dL), and high (>0.3 mg/dL); glucose profile values were applied to calculate the HOMA-IR index, where HOMA-IR = (fasting insulin (µUI/mL) × fasting glucose (mmol/L))/22.5. The possible interference of alcohol consumption and physical activity (both up to 3 times a week) on liver enzyme concentrations and HOMA-IR values was also considered.

### 2.6. Statistical Analysis

To comparatively evaluate diets and sex regarding the measures of interest, we used Linear Models [36], assuming normal distribution. To guarantee the normality assumption, it was necessary to apply the logarithmic transformation to the data. Initially, models with main effects and interactions were tested and reduced models were estimated when some effects did not appear to be significant. In cases where the tests identified statistically significant differences, linear contrasts were made in order to explore the difference in more detail. In the case of the association between quantitative measures transformed to the logarithmic scale, the interpretation of the parameters of the estimated equations was conducted in a regression model called “log-log”, i.e., each 1% increase in the independent variable causes a percentage increase in the dependent variable, estimated by the equation. Therefore, in a log-log regression model, the interpretation of the association between the two variables is made in terms of percentage variation. Model adjustments were evaluated by visual inspection of residuals. For the qualitative variables, the data were summarized using frequencies and percentages, and the associations with groups were investigated using the chi-square test or Fisher’s exact test, when more appropriate. To explore the difference in more detail, the chi-square partition was used. All results were interpreted using a 5% significance level, and Bonferroni adjustments were considered for multiple comparisons. The calculations were performed using the SAS University Edition (Statistical Analysis System) computer system.

## 3. Results

### 3.1. Omnivorous Men and Women Have a Higher Prevalence of Obesity; Vegan Women Have a Higher Prevalence of Underweight

Our sample was predominantly composed of men and women with normal nutritional status (46.92% and 64.71%, respectively), followed by overweight (36.26% and 19.5%, respectively), obesity (13.74% and 10.35%, respectively), and lower frequencies of low weight (3.08% and 5.45%, respectively). Between genders (Figure 2A), we observed a higher prevalence of overweight and obesity in men than in women (X2 = 54.6009, gl = 3, *p* < 0.0001). Eating groups influenced the nutritional status in men (X2 = 37.5515; df = 9; *p* = 0.0001) and in women (X2 = 47.5992; df = 9; *p* = 0.0001). In men (Figure 2B), the prevalence of obesity was lower in lacto-ovo vegetarians (*p* = 0.0004) and semi-vegetarians (*p* = 0.0006) when compared to omnivores and with no statistically significant difference between vegans and omnivores (*p* = 0.0780). In women (Figure 2C), we observed a higher prevalence of obesity in omnivores when compared to lacto-ovo vegetarians (*p* < 0.0001), semi-vegetarians (*p* = 0.0137), and vegans (*p* = 0.00045). The differences are reflected in a higher frequency of low weight among vegans than those with other types of eating habits.

### 3.2. Alcohol Consumption Did Not Influence Liver Enzyme Levels, as Physical Activity Did Not Influence HOMA-IR Levels

We observed no significant interferences of alcohol consumption on AST, ALT, and GGT concentrations between different eating groups, both in men (F_(1, 402)_ = 3.63, *p* = 0.0576; F_(1, 404)_ = 6.33, *p* = 0.5316; F_(1, 391)_ = 0.26, *p* = 0.6082, respectively), and in women who do not menstruate (F_(1, 209)_ = 0.32, *p* = 0.5711; F_(1, 213)_ = 3.81, *p* = 0.0523; F_(1, 211)_ = 1.17, *p* = 0.2810, respectively) and who were menstruating (F_(1, 657)_ = 0.00, *p* = 0.9565; F_(1, 657)_ = 2.42, *p* = 0.1200; F_(1, 652)_ = 0.72, *p* = 0.3961, respectively). The analyses also did not identify significant interferences of physical activity on the association between HOMA-IR and ALT, both in men (F_(1, 391)_ = 1.09, *p* = 0.2973), and in women who do not menstruate (F_(1, 203)_ = 0.81, *p* = 0.3678) and who were menstruating (F_(1, 622)_ = 0.67, *p* = 0.4150). In both sexes, there was a direct association between the increase in HOMA-IR and increase in ALT (*p* < 0.001), regardless of physical activity.

### 3.3. For All Groups, Increased BMI Was Associated with Worse Metabolism, Men Have Higher Levels of GGT and HOMA-IR Than Women, for Vegan Women, Who Do Not Menstruate, the Higher the BMI, the Lower the ALT

The influence of sex and eating groups on the association between BMI values and hs-CRP, HOMA-IR, ALT, AST, GGT, ferritin, and HbA1C values was tested in models with main effects and first- and second-order interactions. The association with BMI was influenced by sex for the variables HOMA-IR (men > women; F_(2, 1220)_ = 4.07, *p* = 0.0173) and GGT (men > women; F_(2, 1254)_ = 5.35, *p* = 0.0048), eating groups for the variable GGT (vegans < other eating groups; F_(3, 1254)_ = 2.75, *p* = 0.0413), and the interaction of sex and eating groups for the variable ALT (F_(6, 1274)_ = 2.17, *p* = 0.0431). In models adjusted separately for sex/menstrual blood loss, including dietary groups as a factor, we observed direct relationships between the progression of BMI values and progression of HOMA-IR values (*p* < 0.0001), GGT (*p* < 0.05), and ALT (*p* < 0.05), for both sexes/menstrual blood loss. Only for women who do not menstruate was the association between BMI and ALT influenced by eating groups (F_(3, 213)_ = 4.07, *p* = 0.0078), whereas for vegans, the associations were in the opposite direction when compared to other eating groups (F_(2, 213)_ = 6.77, *p* = 0.0099), that is, only for vegans, the higher the BMI, the lower the ALT value (Figure 3). A similar inversion was observed in the association between BMI and GGT for these women who do not menstruate; however, it is important to note that the number of vegan women who do not menstruate is small (*n* = 15). The estimated HOMA-IR, GGT, and ALT increment values for each 1% increment in BMI values are described in Table 1.

The association between BMI and the other variables was not influenced by sex and eating groups or by the interaction of these variables (*p* = 0.05). We observed direct associations between BMI values and hs-CRP values (F_(1,1151)_ = 103.90, *p* < 0.0001), AST (F_(1,1268)_ = 5.31, *p* = 0.0214), ferritin (F_(1,1286)_ = 6.93, *p* = 0.0086), and HbA1C (F_(1,1262)_ = 15.97, *p* < 0.0001), regardless of sex, menstrual blood loss, eating groups, and their interactions (Figure 4). For each 1% increase in BMI values, there was an estimated increase of 2.76% in hs-CRP, 0.30% in AST, 1.75% in ferritin, and 0.09% in HbA1C.

### 3.4. In Obesity, Vegetarian Individuals Have a Lower Elevation of GGT and Ferritin

Although previous analyses have suggested that the association between hs-CRP, HOMA-IR, ALT, AST, GGT, ferritin, and HbA1C values with BMI were not different among different dietary groups, we sought to assess the effect of dietary groups considering only the 153 obese individuals (BMI ≥ 30 kg/m^2^). For this analysis, we compared omnivores and vegetarians (lacto-ovo-vegetarians, semi-vegetarians, and vegans grouped together). These analyses were conducted with the aforementioned transformed variables, in order to ensure that the normality assumption of the models were valid. For obese individuals, there was an effect of dietary groups on GGT and ferritin concentrations, regardless of sex and menstrual blood loss: vegetarian individuals had lower values of these variables than omnivorous individuals (Figure 5 and Appendix A).

## 4. Discussion

The beneficial effect of adopting a well-planned vegetarian diet is fully recognized in the scientific literature [37], including showing that individuals with free food intake have a greater effect on body weight reduction when they are vegetarians (mainly vegans), compared with omnivores [38,39,40].

A systematic review concluded that there is robust evidence that the adoption of vegetarian habits can bring benefits in the medium term (=24 months) on body weight, energy metabolism, and systemic inflammation in healthy, obese individuals with type 2 diabetes mellitus (DM2), compared to omnivorous habits [41].

Based on this prior knowledge, we decided to verify if there are also metabolic differences between vegetarians and omnivores according to their body weight, information that has been absent in the literature so far.

In the selection of individuals, we accepted those who used alcoholic beverages and practiced physical activity. As alcohol can influence ALT, AST, and GGT levels, and as physical activity can influence insulin sensitivity, we proceeded to evaluate the individuals in relation to these variables. We did not find any interference from these factors in this regard, possibly because our inclusion criteria limited the use of alcohol and the practice of physical activity up to 3 times a week, excluding alcoholics and athletes from the sample. Therefore, all selected individuals were included in our analyses.

We observed that omnivorous men and women have a higher prevalence of obesity when compared to vegetarians. Vegan women in our study showed a higher prevalence of underweight compared to the other groups. These data show the importance of separating, in population studies that aim to make biochemical or outcome comparisons, individuals with different BMI, as excess weight causes metabolic changes and, considering only the food option (omnivores or vegetarians), may not represent the effect resulting solely from the diet when disregarding the degree of body adiposity of each individual.

Obesity is determined by high BMI values (≥30 kg/m^2^), which promote systemic inflammation and, consequently, the development of hepatic steatosis and insulin resistance (IR) [42]. Systemic inflammation may be reflected by increased circulating concentrations of ferritin, an acute phase protein [42]. Accordingly, elevated ferritin levels are associated with increased hepatic steatosis and ALT [42] and an increased risk of developing type 2 diabetes in both eastern and western populations [43]. Considering that sex and blood loss can influence systemic ferritin levels [44,45,46], we chose to study this association separately in non-menstruating and menstruating women and men. In addition, the comparison between eating groups included analyses in the general population and separately in obese individuals.

Our study showed that for men and women (with or without blood loss), both vegetarians and omnivores, the greater the body mass index, the worse the metabolic conditions. In particular, this change in nutritional status was associated with an increase in IR and GGT, both of which were more pronounced in men than in women. In the general sample, when the body mass index increases, the levels of hs-CRP, ALT, AST, Ferritin and HbA1C increase.

HOMA-IR was analyzed as a marker of IR and GGT as a marker of canalicular dysfunction and oxidative status, which can be intensified by increasing body weight. Although the direct association between GGT and BMI has indicated an initial influence of dietary group (vegans < other groups), the latter was not confirmed in the final adjustment models.

Vegetarian habits are associated with better glycemic control [24,27,47,48], but our data indicate that in the condition of overweight and obesity, the IR increases, both in vegetarians and in omnivores. It is possible that the stronger association between BMI and HOMA-IR observed in men was due to the greater accumulation of visceral than peripheral fat, but these data were not evaluated in our study.

ALT was analyzed as a marker of liver damage. We found a direct association between the increase in ALT concentrations and BMI values in men and women, but among women who do not menstruate, this was influenced by dietary groups: it was lower in vegans when compared to other dietary groups. We raised some hypotheses for this finding, such as the possibility that it is associated with a lower consumption of saturated fat and more fiber, changing the microbiota in vegan individuals. These changes can keep the intestinal barrier intact and prevent endotoxemia and its inflammatory side-effects such as liver damage [49,50,51,52]. Assessment of the dietary intake and gut microbiota composition could confirm whether saturated fat intake or microbiome diversity influenced the direct association between ALT and BMI and explain why this was not observed in men and women who menstruate but were not made in our study. Likewise, this hypothetical change in dietary profile and microbiome should also cause differences in men and women who menstruate, but it did not. Therefore, we have no plausible explanation for this finding in our study.

In overweight, we observed no significant difference in relation to the progressive increase in the variables, but for each 1% increase in BMI, we saw that the vegan diet presented a smaller percentage (nonsignificant) increase in the parameters evaluated. Therefore, we proceeded with the evaluation of only obese individuals who, due to the sample size, needed to group semi-vegetarians, lacto-ovo, and vegans within one group, which was called vegetarian, to be compared with omnivores.

In the obese condition, vegetarians showed a lower elevation of GGT and ferritin, regardless of sex and menstrual blood loss, which suggests a potential protection of this dietary habit against morbidities associated with excess weight. Therefore, we continue our discussion regarding ferritin and GGT.

It is known that there is a direct correlation between IR and circulating levels of ferritin. A study that evaluated a sample of 12,090 individuals (6378 men and 5712 women) with type 2 diabetes, altered fasting glucose, and euglycemia reported that circulating levels of ferritin decrease as the degree of IR decreases. Adjustment of variables by multiple regression analysis showed that IR was independently associated with ferritin concentration in men but not in women [53]. In addition to the correlation between IR and increased ferritin, there seems to be a causal association between ferritin concentrations and insulin sensitivity. Hua et al. reported greater insulin sensitivity and lower mean ferritin concentration in lacto-ovo vegetarians than in omnivores and showed that, after therapeutic bleeding to achieve similar ferritin levels to vegetarians, omnivores showed a 40% increase in insulin sensitivity [54]. Accordingly, another study showed that lower ferritin levels are associated with better glycemic control and protection against DM 2 [55].

Our group recently published data from the same sample presented in this article regarding iron metabolism, confirming the association between increased ferritin levels and IR and inflammation (assessed by hs-CRP dosage) [21]. Omnivorous individuals in our sample had significantly higher ferritin levels than vegetarians.

A recent study evaluated, by means of nuclear magnetic resonance and liver biopsy, the levels of liver fat and correlated it with serum ferritin levels in 167 individuals. The results showed that the increase in hepatic fat content correlates with the increase in ferritin levels [56]. However, it is not reliable to be used alone as a marker of liver fibrosis [57].

Although GGT is widely used in liver assessment, it is known that its main function is to metabolize reduced extracellular glutathione (GSH), allowing its precursor amino acids to be assimilated and reused for intracellular synthesis [9], as illustrated in the introduction to this article. In this context, elevated GGT is an indicator of glutathione depletion (especially in the liver) [58], that is, the higher the level of GGT measured in the blood, the greater the need for intracellular glutathione in the absence of cholestasis. Small elevations in the levels of this enzyme affect metabolism. A systematic review and meta-analysis showed that for every 5 U/L increase in serum GGT levels, there is a relative risk of 1.08 (95% CI: 1.04–1.13) of developing systemic arterial hypertension and 1.04 (95% CI: 1.03–1.05) of developing cancer in general [59]. In addition, GGT levels >24 U/L are associated with a 22% increase in all-cause death compared to GGT levels <24 U/L, and the cardiovascular risk is 67% higher in individuals with pre-existing cardiovascular disease, particularly in men who have already had an acute myocardial infarction [60]. GGT is an early marker of obesity development, even after adjusting for other variables [61].

Systematic reviews and meta-analyses associate an elevation of GGT, regardless of alcohol consumption, with an increased risk of mortality from cardiovascular disease and all-cause death [62,63], systemic arterial hypertension [14], metabolic syndrome [15,18], stroke [13], cancer [59], and diabetes [16].

Elevated serum ferritin levels with a concomitant elevation of GGT indicate increased hepatic iron overload [58]. There are a few articles published in the literature that analyzed the concomitant increase in ferritin and GGT. However, it is known that GGT interacts directly and indirectly with iron, with the first report of this interaction described by Brown et al. (1998) [64], based on animal studies. GGT induces oxidative stress in the arterial wall in the presence of free iron, in addition to damaging erythrocyte membranes, releasing metals into the bloodstream [58]. Elevated levels of GGT indicate increased oxidative activity, especially in the presence of iron or copper [58].

The pro-oxidative property of the interaction of GGT with iron has been one of the theoretical bases that justify observations in studies, given the potential synergistic association with increased IR [65] and the development of DM2 [66], metabolic syndrome [67], and the development of chronic kidney disease [68]. Accordingly, one of these studies found in its analysis an association between concomitant elevated levels of ferritin and GGT with higher levels of plasma malondialdehyde, which has a supporting action in inflammation by stimulating the production of pro-inflammatory cytokines. Malondialdehyde is an end product of lipid peroxidation, associated with increased oxidative stress [66].

The Framingham Heart Study followed 3451 participants and demonstrated that the increase in serum GGT predicts metabolic syndrome, cardiovascular events, and death, suggesting that it is used as a marker of cardiac risk and metabolic alterations [69].

The studies that evaluated the association between GGT and ferritin show strong results on the metabolic risk resulting from the joint increase in these markers. There are three publications with the same sample of patients (1024 participants, 436 men and 588 women) where the authors divided the values obtained for GGT and ferritin into three groups: Group 1 (GGT and ferritin not reaching the fourth quartile), Group 2 (GGT or ferritin in the fourth quartile), and Group 3 (GGT and ferritin in the fourth quartile).

The first publication evaluated the association of GGT and ferritin with DM2. Group 3 had the most abnormalities in glucose metabolism and the highest risk of DM2 (regardless of age, BMI, and family history of DM2) and the highest levels of plasma malondialdehyde. The authors concluded that GGT and ferritin were correlated and had a synergistic effect on the risk of T2DM, probably due to increased oxidative stress [66].

The second publication showed that the risk of metabolic syndrome was significantly higher in women who had a joint elevation of GGT and ferritin. The authors suggested that GGT and ferritin synergistically correlate with the risk of metabolic syndrome, and that they could potentially be used as predictive biomarkers for this syndrome [67].

The third publication investigated the effect of GGT and ferritin on chronic kidney disease. Individuals with GGT or ferritin in the highest quartiles had a greater association with the risk of chronic kidney disease. With the increase in GGT and ferritin levels, we observed an increase in albuminuria and malondialdehyde levels. In group 3 (GGT and ferritin in the fourth quartile), the increased risk was independent of age, BMI, alcohol use and presence of diabetes, hypertension, hypertriglyceridemia, and metabolic syndrome [68].

Wei et al. evaluated 756 individuals aged 20 to 74 years and submitted them to an oral glucose tolerance test, in addition to measuring GGT and ferritin levels. We observed a positive correlation between HOMA-IR and GGT or ferritin after adjustment for multiple confounders. The authors concluded that insulin resistance was associated with an increase in GGT and ferritin, and the joint assessment of these two elements can better predict patients who have IR [65].

More recently, a study evaluated the association between GGT and ferritin in 77 individuals with metabolic syndrome and 77 controls and found a positive association between the increase in the levels of these two markers with the increase in abdominal circumference and serum triglyceride levels. The authors suggest the use of these markers in the early assessment of individuals with metabolic syndrome and point to a positive correlation between GGT and triglycerides, indicating their role in oxidative stress, leading to inflammation and the development of the metabolic syndrome [70].

As elevated ferritin is an element indicative of greater metabolic oxidative action and high GGT is indicative of a reduced antioxidant effect, this joint elevation may predispose the individual to a risky metabolic condition. Thus, our observation that obese vegetarian individuals had lower circulating levels of ferritin and GGT than omnivores suggests that the vegetarian diet may attenuate their metabolic disturbances.

Looking at GGT as the most protective antioxidant element in the vegetarian group has support in the scientific literature also from the food point of view. The evaluation of the antioxidant content of more than 3100 foods showed that those from the plant kingdom have 64.27 times more antioxidants than the animal kingdom per 100 g of product [71]. In fact, several studies have shown a higher antioxidant intake and better metabolic antioxidant response in vegetarian subjects than in omnivorous subjects [72,73,74,75]. Taken together, these observations suggest that a vegetarian diet can help prevent the pro-oxidative action of GGT through the availability of free iron, as it has a potentially greater amount of antioxidants in its composition than the omnivorous diet.

In addition, there are reports that meat, fish, dairy products, and fruits are the main foods contaminated with organochlorines and that foods of animal origin are more contaminated by several other types of pesticides than those of plant origin [76,77,78,79]. The use of organic foods contributes to the lower intake of nonorganochlorine pesticides in vegetarian groups [80,81]. Additionally, studies show that vegetarian individuals have a much lower mercury contamination when compared to omnivores [82]. The circulating increase in GGT may be associated with increased exposure to environmental xenobiotics. It turns out that this liver enzyme is needed to metabolize xenobiotics in the liver and other organs, including the lungs [58]. Therefore, it is also possible that, in obese individuals, vegetarian habits may also help to prevent the elevation of GGT, by presenting not only more antioxidants, but also potentially lower amounts of xenobiotics in their composition than the omnivorous diet.

Our main limitation is the fact that we did not assess food consumption and composition of the microbiota of the population studied, which could help understand whether macronutrient intake, dietary fat profile, fiber content, and intestinal microbiota influenced the association between ALT and BMI, and why this was observed only in women who do not menstruate. A main asset is that we have the selection of participants, with an assessment of the possible interference of alcohol consumption and physical activity on liver enzyme levels and insulin sensitivity, respectively, in order to eliminate confounding variables from our sample. The large sample size was also one of the strengths of our study, as well as the inclusion of individuals of both sexes and different dietary profiles. This feature allowed the comparison between men and women, as well as the control of menstrual blood loss in women while maintaining a significant sample size. This is the first publication available in the literature with such comparative data between vegetarians and omnivores.

## 5. Conclusions

Both vegetarians and omnivores show worse metabolic parameters when having a higher body mass index. However, regarding obesity, vegetarians showed a better antioxidant status (lower GGT elevation) and lower inflammatory status (lower ferritin elevation), which may provide them with potential protection in the development of morbidities associated with obesity.

## Figures and Tables

**Figure 1 nutrients-14-02204-f001:**
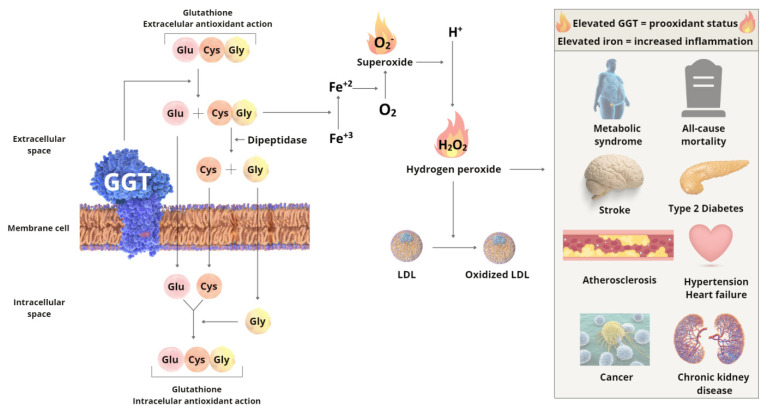
Correlation between the prooxidant effect of gamma-glutamyl transferase (GGT), iron, and disease. Cysteine (Cys); Glutamic acid (Glu); Glycine (Gly); Low-density lipoprotein (LDL); Iron (Fe). Adapted from Ndrepepa, G. and A. Kastrati [10].

**Figure 2 nutrients-14-02204-f002:**
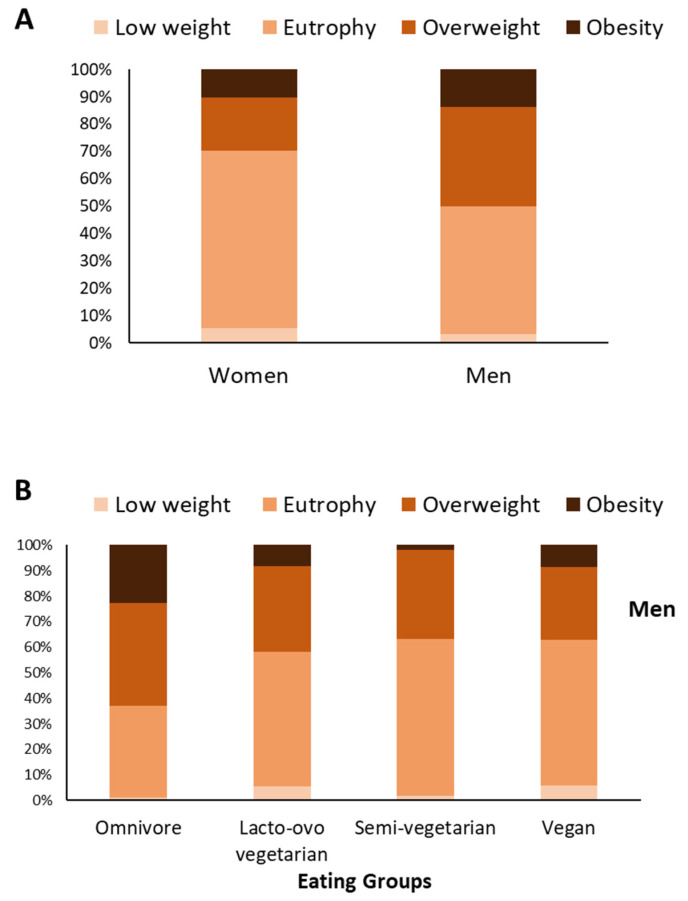
Frequency of low weight, normal weight, overweight, and obesity, according to sex and eating habits. The figure shows the frequency of distribution of nutritional status between men and women (**A**) and between eating groups in men (**B**) and women (**C**). The diagnosis of nutritional status was based on the values of body mass index (BMI): <18.5 = low weight; ≥18.5 and <25.0 = normal weight; ≥25.0 and <30.0 = overweight; ≥30.0 = obesity. Data expressed in percentage.

**Figure 3 nutrients-14-02204-f003:**
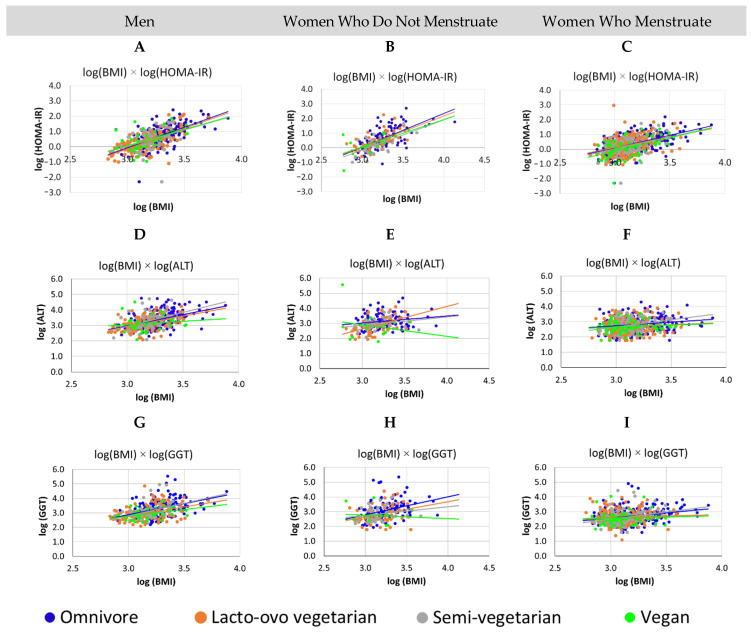
Associations between body mass index (BMI) values and Homeostatic Model Assessment-Insulin Resistance (HOMA-IR), alanine aminotransferase (ALT), and gamma glutamyl transferase (GGT) values in men and women who do not menstruate and who menstruate with different eating groups. The figure shows a direct association between BMI values and HOMA-IR, ALT, and GGT values in men (**A**,**D**,**G**, respectively) and in non-menstruating (**B**,**E**,**H**, respectively) and menstruating (**C**) women (**F**,**I**, respectively), regardless of eating groups. The exception was among women who do not menstruate, where vegans showed an inverse association between ALT and BMI, compared to other dietary groups (*p* = 0.0099).

**Figure 4 nutrients-14-02204-f004:**
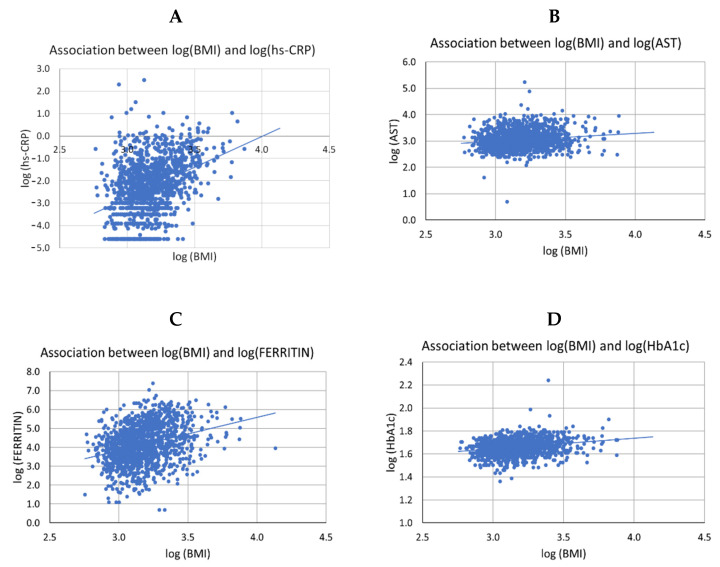
Associations between body mass index (BMI) values and C-reactive protein (hs-CRP), aspartate aminotransferase (AST), ferritin, and glycated hemoglobin (HbA1C) values, regardless of sex/menstrual loss of blood and eating groups and their interactions. The figure shows a direct association between BMI values and CRP (**A**), AST (**B**), ferritin (**C**), and HbA1C (**D**) values, regardless of sex, menstrual blood loss, eating groups, and their interactions (*p* = 0.0500).

**Figure 5 nutrients-14-02204-f005:**
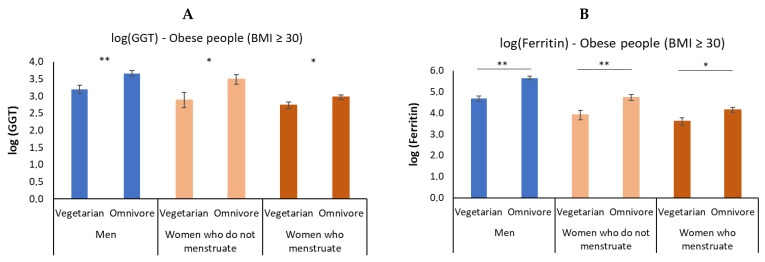
Means and standard errors of gamma glutamyl transferase (GGT) and ferritin according to dietary groups (omnivore and vegetarian) in men and women who menstruate and who do not menstruate with obesity (BMI ≥ 30 kg/m^2^). The figure shows lower levels of GGT and ferritin in vegetarian subjects than in omnivorous subjects, regardless of sex and menstrual blood loss (*p* = 0.0395)—* *p* < 0.05, ***p* < 0.01. (**A**) GGT according to sex and dietary group; (**B**) Ferritin according to sex and dietary group.

**Table 1 nutrients-14-02204-t001:** Estimates of percentage increment of HOMA-IR, GGT, and ALT for each 1% increment of BMI values. The last line of the table presents the *p*-value of the comparison of eating groups regarding the slopes of the association of each variable with BMI.

Eating Habit	Men			Women Who Do Not Menstruate			Women Who Menstruate		
	HOMA-IR	GGT	ALT	HOMA-IR	GGT	ALT	HOMA-IR	GGT	ALT
ONV	2.70%	1.53%	1.45%	2.30%	1.22%	0.47%	1.70%	0.70%	0.47%
LOV	2.67%	1.24%	1.35%	2.11%	1.02%	1.50%	1.45%	0.22%	0.30%
SMV	2.16%	1.73%	1.79%	2.05%	0.58%	0.55%	1.63%	0.94%	0.92%
VGN	2.13%	0.90%	0.43%	1.86%	−0.23%	−0.77%	1.78%	0.28%	0.25%
*p* Value	0.7349	0.6113	0.2198	0.9119	0.2744	0.0078 *	0.8428	0.0985	0.3325

LOV, lacto-ovo vegetarian; ONV, omnivore; SMV, semi-vegetarian; VGN, vegan. * The specific comparison between vegans vs. other eating groups resulted in a *p* = 0.0099.

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
