# Peer review of "Obese Vegetarians and Omnivores Show Different Metabolic Changes: Analysis of 1340 Individuals"

_nutrients, 2022, doi:10.3390/nu14112204_

Round 1

Reviewer 1 Report

Introduction: 

Remove Figure 1 from the manuscript.  It is not referred to anywhere in the Results and Discussion, so i question why it is included in your paper.

Materials and Methods:

Unifesp/EPM Write out on words instead of using an acronym.

"Model adjustments were made by visual inspection of residual:.  Statement appears to be very subjective.  What criteria were used to make model adjustments from visual inspection of the residual?

The authors state "To guarantee the normality assumption it was necessary to apply the logarithmic transformations to the data" I question the necessity of making all these logarithmic transformations.

Results

I think readers will have trouble interpreting the meaning of the many log vs log correlation graphs presented in the paper.

In my judgement, the manuscript is too long.  So much data is presented and talked about in the text, that I am worried that readers will lose sight of the key findings of the study.

Reviewer 2 Report

This great study shows the difference in systemic inflammatory parameters between various well-defined and to-the-point groups. The paper is wel worked out and hypothesis-generating. I have some remarks:

  • Could the negative effects of omnivore diets be due to the actual processing of meat, rather than to the meat itself? And is there a difference between "red" meat and "white" meat?
  • I am surprised to notice no effect at all of physical activity (even though the included patients were non-athletes)? Do the authors have an explanation for this?
